# Fisher zeroes and the fluctuations of the spectral form factor of chaotic systems

**Guy Bunin[1], Laura Foini[2⋆] and Jorge Kurchan[3]**

**1** Department of Physics, Technion-Israel Institute of Technology, Haifa 32000, Israel
**2** IPhT, CNRS, CEA, Université Paris Saclay, 91191 Gif-sur-Yvette, France
**3** Laboratoire de Physique de l'École Normale Supérieure, ENS, Université PSL,
CNRS, Sorbonne Université, Université de Paris, F-75005 Paris, France

⋆ laura.foini@ipht.fr

## Abstract

The spectral form factor of quantum chaotic systems has the familiar 'ramp + plateau' form. Techniques to determine its form in the semiclassical [1] or the thermodynamic [2] limit have been devised, in both cases based on the average over an energy range or an ensemble of systems. For a single instance, fluctuations are large, do not go away in the limit, and depend on the element of the ensemble itself, thus seeming to question the whole procedure. Considered as the modulus of a partition function in complex inverse temperature $\beta_R + i\beta_I$ ($\beta_I \equiv \tau$ the time), the spectral form factor has regions of Fisher zeroes, the analogue of Yang-Lee zeroes for the complex temperature plane. The large spikes in the spectral form factor are in fact a consequence of near-misses of the line parametrized by $\beta_I$ to these zeroes. The largest spikes are indeed extensive and extremely sensitive to details, but we show that they are both exponentially rare and exponentially thin. Motivated by this, and inspired by the work of Derrida on the Random Energy Model, we study here a modified model of random energy levels in which we introduce level repulsion. We also check that the mechanism giving rise to spikes is the same in the SYK model.

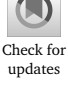

# 1  Introduction

The spectral form factor (SFF) of a quantum system is the energy level density two-point correlation. It has been widely studied in Random Matrix Theory, and in generic quantum chaotic systems [3–12]. More recently [13–16] (see also [17]) it has been revisited in the context of the Sachdev-Ye-Kitaev (SYK) model [18, 19], which is thought to provide a 'toy' version of holography and gravity. The SYK model has random interactions, like a spin glass. Just as with spin glasses, some quantities are self-averaging in the large size limit, their sample-to-sample fluctuations vanishing with some inverse power of the size $N$. It may seem then surprising that, when one plots the spectral form factor of a single realization over an exponentially long range of times, one finds that it contains fluctuations (spikes) that do not vanish upon increasing the system's size. The spikes are extremely sensitive to a perturbation in the couplings, and thus an averaged result seems to be blind to them. Moreover, rather surprisingly, these fluctuations do not seem to contribute to the sample-averaged result, in spite of them being large.

In this paper we show that (at least some of) such fluctuations are the manifestation of the Fisher zeroes of the partition function in complex temperature, each spike is an extremely rare near-miss of the $\beta_R + i\beta_I$ line to one of these zeroes. This mechanism explains why, even if spikes seem large, their averaged contribution to the spectral form factor is in fact exponentially small in $N$. The average density of Fisher zeroes is readily available in an averaged, large $N$ computation, and gives us direct access to the density of spikes. Note also that although sometimes these spikes are attributed to the discreteness of the spectrum, regions with finite densities of zeros are found in classical statistical models with continuous degrees of freedom [20].

Some thirty years ago, Derrida [21] computed the partition function in complex temperature and the distribution of Fisher zeroes for the simplest spin glass, the Random Energy Model (REM) [22], a system of Poisson-distributed, independent energy levels. The spectral factor turns out to have a 'slope', and a 'plateau', but no ramp. The 'plateau' occurs in a region of the complex temperature plane where there is a surface density of zeroes. In that phase, random fluctuations destroy the delicate coherence needed for analytic continuation to work. This is the interesting phase for our purposes here.

Our main task will be to make the analog of the REM but with level-repulsion, just by transing the spectrum obtained from a random matrix to any desired level density, or equivalently, of entropy $s(e)$. Such a model plays the role of a 'null model', having just the bare minimum level structure compatible with it being chaotic.

The structure of the manuscript is as follows: in Section 2 we discuss the general properties of Fisher zeros and their effect on the SFF. In Section 3 we review the phase diagram of the REM in complex temperature. In Section 4 we modify the REM to include level repulsion and we compute the SFF of this model.

# 2  Fisher zeroes and near-missed zeros

The spectral factor of a Hamiltonian at $\beta = \beta_R + i\beta_I$ (here $\beta_I \equiv \tau$, the time), is defined as $|Z(\beta)|^2$, where and $Z(\beta)$ is the partition function at complex temperature $\beta$:

$$Z(\beta) = \sum_i e^{-\beta E_i}, \qquad (1)$$

$E_i = Ne_i$, we use lower case for intensive quantities. For finite number $N$ of degrees of freedom, $Z$ is an analytic function of $\beta$, with isolated zeroes. The set of zeroes in complex temperature,

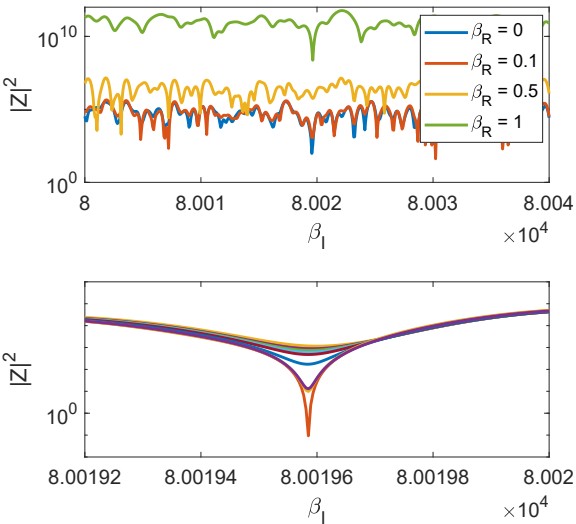

Figure 1: Upper panel: logarithmic plot of $|Z(\beta_R + i\beta_I)|^2$ vs $\beta_I$ in the plateau regime for a model of random energy levels with level repulsion and Gaussian density of states. Lower panel: zooming near a spike, and identifying it as a near-miss to a zero by changing the value of $\beta_R$ slightly.

analogous to the Yang-Lee zeroes in complex magnetic field, are referred to as Fisher zeroes [23, 24, 24, 25]. As we go to the thermodynamic limit, these zeroes may assemble into lines with a finite linear density, or areas of constant surface density, or even fractal distributions [20, 21, 26–28]. The density of zeroes is obtained directly from $\ln|Z|$ as:

$$\Omega(\beta_R, \beta_I) = \frac{1}{2\pi}\left(\frac{\partial^2}{\partial\beta_R^2} + \frac{\partial^2}{\partial\beta_I^2}\right)\frac{1}{N}\ln|Z|. \tag{2}$$

In Eq. (2) we can interpret $\frac{1}{N}\ln|Z|$ as the electrostatic potential in two dimensions generated by the distribution of charges $\Omega(\beta_R, \beta_I)$.

It may seem paradoxical that $\frac{1}{N}\ln|Z|$ can have a smooth thermodynamic limit, and at the same time have a density of infinitely deep spikes in the zeroes of $Z$. The situation is in fact best understood by analogy with a charged surface. We know that there is meaningful smooth electric potential on the surface and outside, and yet we also know that near each charge the potential has a spike. Somehow the coarse-grained limit recovers the smoothness, and makes the spikes go away. This is possible because the regions in the surface around charges where the discontinuity is appreciable is small compared to the total surface.

In the models we shall discuss below, something similar happens for the complex temperature plane. We shall show that one has regions with an $O(N)$ number of zeroes per element $d\beta_R d\beta_I$, and an averaged, smooth extensive value for $\ln|Z(\beta)|$. There are, however, small regions around each zero where $\ln|Z|$ has a spike. As an example, in Figure 1 we choose a large spike, and locate explicitly the nearby pole responsible for it.

Consider one such region and compute $\ln|Z(\beta_R + i\beta_I)|$ along a trajectory of fixed $\beta_R$ and a large range of $\beta_I$ traversing it. The probability that the trajectory actually hits a zero is either zero or becomes zero upon small perturbations of the Hamiltonian.[1] A near-miss along the

---

[1]An exception to this is when there is a symmetry (for example along the imaginary axis), then one may expect a density of zeroes lying *exactly* on the invariant lines. See [29].)

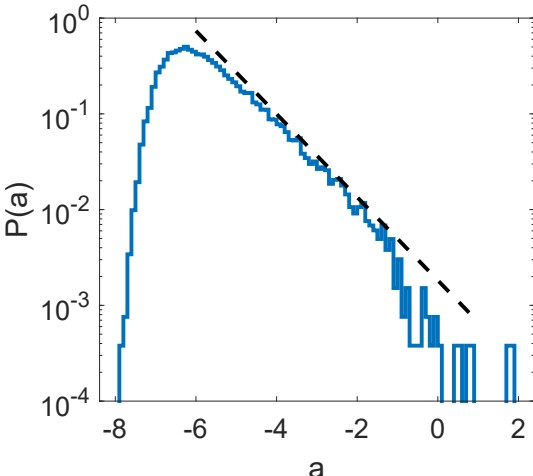

Figure 2: The large-deviation function of spike depths, $d$, along a line with constant $\beta_R = 0.1$ and varying $\beta_I$ inside the plateau, in a simulation of the model with level repulsion and $N = 16$. The dashed trend line is exponential, $\exp(-d)$.

line $\beta_R = const$ of a zero just beside it, at $(\beta_R + e^{-aN}, \tau_0)$ gives

$$|Z(\beta)| = \sqrt{e^{-2Na} + (\beta_I - \tau_0)^2}\, e^{N G_\infty(\beta)}\,, ^2 \tag{3}$$

where $G_\infty$ is analytical in a domain around $(\beta_R, \tau_0)$. Then, $\log|Z(\beta)|/N$ has a spike of negative dip $= -a$. The largest spike in the factor along a trajectory that is exponentially long in $\tau = \beta_I$ is estimated from the closest near miss. Let us say we look at time-intervals of order $\Delta\beta_I = e^{bN}$ for spikes of magnitude $\frac{1}{N}\ln|Z| \sim a$.

If the average surface density of zeroes along such a line is $\bar\rho$ (which may be even $1/N$), then the number of such spikes is: $\mathcal{N}(a) \sim \bar\rho e^{(b-a)N}$ and the *large deviation function* of spike sizes is:

$$\frac{1}{N}\log\mathcal{N}(a) \sim b - a + \frac{1}{N}\log\bar\rho\,. \tag{4}$$

The largest spike found in $\Delta\beta_I$ is thus of order $a \sim b$. Figure 2 confirms this asymptotic law for a model with level repulsion we shall discuss below.

Are the positions of spikes truly independent? This depends on the correlations of the zeroes involved. We know that generically the zeroes of a random polynomial repel one another [29], but they could be, for non-random situations, some other correlations. On the other hand, asking for the closest misses of a trajectory could force these to be far away, much in the same way that the Grad limit of Boltzmann's equation makes collisions independent [30].

If these spikes associated to near misses of zeroes are the only ones that survive in the thermodynamic limit, then the fact that they are exponentially rare in time $\tau \equiv \beta_I$ means that their averaged contribution to the logarithm of the spectral form factor will be negligible to all orders in $1/N$.

In Fig. 3 we show that the same mechanism is at play in the SYK model. The model is defined by the Hamiltonian:

$$H = \sum_{1 \leq i \leq j \leq k \leq l}^{N} J_{ijkl}\, \chi_i \chi_j \chi_k \chi_l\,, \tag{5}$$

with $\chi_i$ $N$ Majorana fermions and $J_{ijkl}$ random i.i.d. gaussian numbers [18,31]. In Fig. 3 we show a zoom of the SFF for the SYK which aims at highlighting the presence of a near missed zero upon varying $\beta_R$.

---

[2]To see this properly one should study the Weierstrass product development of $Z(\beta)$ in the vicinity of one zero.

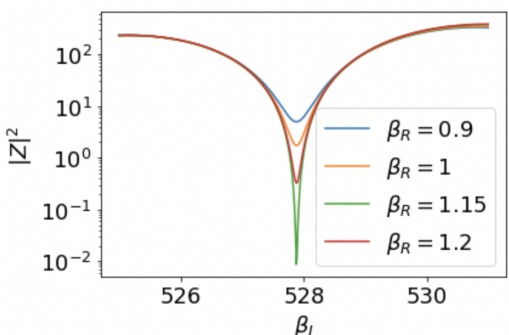

Figure 3: Same as Fig. 1 for a fluctuation of SYK model. Here too, the presence of a nearby Fisher zero is seen as the cause of the big spike.

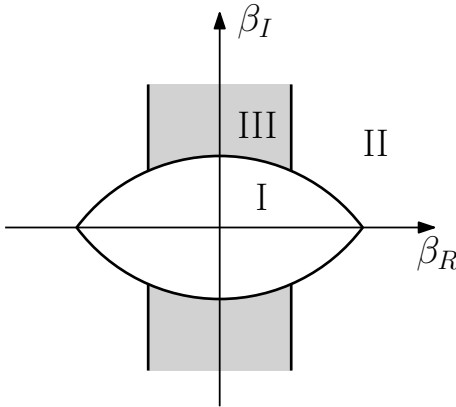

Figure 4: Complex temperature phase diagram of the (Poissonian) Random Energy Model as in [21]. Phase II, the spin glass phase, does not exist for models like SYK, and is mostly irrelevant here.

## 3 The random energy model

The structure of the zeroes is very well understood in the case of the Random Energy Model (REM) studied by Derrida [21] years ago. The REM is characterized by $2^N$ independent energy levels drawn from a Gaussian distribution with mean zero and variance $\langle E^2 \rangle = \frac{N}{2}$. The average density of energy levels is given by:

$$\langle \rho(E) \rangle \propto e^{N\left[\log 2 - \left(\frac{E}{N}\right)^2\right]} = e^{Ns(e)} \tag{6}$$

(the normalization is such that $\int \mathrm{d}e \langle \rho(Ne) \rangle \simeq 2^N$), and this should be used to compute the partition function

$$Z(\beta) \simeq \int \mathrm{d}e\, \rho(Ne) e^{-N\beta e}, \tag{7}$$

in the regime $|e| = |\frac{E}{N}| \geq \sqrt{\log 2} = -e_c$. Beyond this value there are no levels and the density in (7) should be taken as zero.

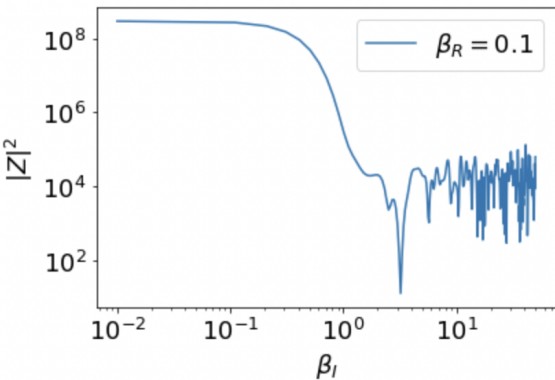

Figure 5: A plot along a vertical line through the gray region (phase III) of Fig. 4 of the Poissonian REM model. There is the 'slope' regime and the 'plateau' regime, but no 'ramp'. The fluctuations in the plateau are large. Here $N = 14$ and $\beta_R = 0.1$.

In the complex temperature plane one can recognize three different phases:

*Analytic (liquid) phase I.* The average of $Z$ itself is well-defined for large $N$:

$$\langle Z(\beta) \rangle \simeq \int de \, \langle \rho(Ne) \rangle e^{-N\beta e} = \int_{-e_c}^{e_c} de \, e^{N\left[\ln 2 - e^2 - \beta e\right]} \,. \tag{8}$$

This may be evaluated for $|\beta| = O(1)$ by saddle point (the liquid phase), $2E_{sp} = -N\beta$ and we get:

$$\langle Z(\beta) \rangle \sim 2^N e^{N\frac{\beta^2}{4}} \,, \qquad \frac{1}{N} \log |Z_{typ}(\beta)| \sim \log 2 + \frac{\beta_R^2 - \beta_I^2}{4} \,, \tag{9}$$

and $Z_{typ}$ describes the typical value of a random set of energy levels.

*Analytic (glass) phase II.* Here too the limit $\lim_{N \to \infty} \frac{1}{N} \ln Z$ is well defined. However, at sufficiently low (real) temperatures, there is no saddle point in a region with positive density of levels, so we conclude that the system gets frozen around the lowest energy density:

$$\frac{1}{N} \log Z_{typ}(\beta) = \beta e_c \,, \tag{10}$$

at the critical temperature at which

$$\beta_c = \left. \frac{ds}{de} \right|_{e_c} \,. \tag{11}$$

Note that in a model with $\lim_{e \to 0} s'(e) = \infty$, the transition happens at zero temperature, as in SYK.

Although this reasoning is for real $\beta$, there is a region in the complex plane (to be determined) where the solution may be continued.

*Non-analytic phase III.* Here a new phenomenon appears. As one analytically continues solutions to larger and larger $\beta_I$ the integrand becomes more oscillating and its value becomes exponentially smaller due to precise cancellations of semi-cycles of $e^{-i\beta_I E}$. These cancellations are however imperfect, because the density of levels is in fact a random function. For large enough $\beta_I$ the fluctuations of the integrand destroy the coherence of integral, and fluctuations themselves dominate. To estimate this, one has to compute [21] the fluctuating part of the partition function. This can be inferred from correlation between energy levels:

$$\langle \delta\rho(E)\delta\rho(E') \rangle = \langle \rho(E)\rho(E') \rangle - \langle \rho(E) \rangle \langle \rho(E') \rangle = \langle \rho(E) \rangle \delta(E - E') \,, \tag{12}$$

as befits a Poissonian distribution. Putting $Ne_+ = \frac{1}{2}(E + E')$ and $Ne_- = E - E'$ one has

$$\langle |Z(\beta)|^2 \rangle = \int \mathrm{d}e\,\mathrm{d}e' \langle \delta\rho(Ne')\delta\rho(Ne) \rangle e^{-\beta_R N(e+e')-i\beta_i N(e-e')}$$

$$= \int_{-e_c}^{e_c} \mathrm{d}e_+ \rho(Ne_+)e^{-N2\beta_R e_+} \int \mathrm{d}e_- \delta(Ne_-)e^{-iN\beta_i e_-}. \tag{13}$$

The saddle point evaluation gives $e_{sp} = -\beta_R$ and

$$\langle |Z(\beta)|^2 \rangle \sim e^{N\ln 2 + N\beta_R^2} = \langle |Z(2\beta_R)| \rangle, \tag{14}$$

$$\frac{1}{N}\log |Z(\beta)|_{typ} \sim \frac{\log 2}{2} + \frac{1}{2}\beta_R^2. \tag{15}$$

One is thus left with the task of comparing three $|Z_{typ}|$, the transition lines are obtained equating Eqs. (9), (10) and (14):

$$\frac{1}{N}\log|Z_I| = \log 2 + \frac{\beta_R^2 - \beta_I^2}{4},$$

$$\frac{1}{N}\log|Z_{II}| = \beta_R e_c,$$

$$\frac{1}{N}\log|Z_{III}| = \frac{\log 2}{2} + \frac{1}{2}\beta_R^2. \tag{16}$$

The phase diagram obtained in Ref [21] is shown in Figure 4.

The density of zeroes, normalized with $N$ is zero in phases I and II, as is to be expected, since the partition function is analytic there. In phase III on the contrary we get:

$$\Omega(\beta_R, \beta_I) = \frac{1}{2\pi}, \tag{17}$$

i.e. there is a constant surface density of zeroes. There are also linear densities in the boundaries I-II and I-III. In Figure 5 we show a plot of $\ln|Z|$ along a line of constant $\beta_R$: there is a 'slope' region, corresponding to phase I, and a 'plateau' region (phase III) with negative spikes associated with near-misses of zeroes.

## Comments on annealed and quenched averages

In the three cases above, the task is to compute the large $N$ limit of

(a) $\frac{1}{N}\langle \log Z(\beta) \rangle$,    if the large $N$ limit exists,

(b) $\frac{1}{N}\langle \log |Z(\beta)| \rangle = \lim_{n\to 0} \frac{1}{2}\frac{d}{dn}\langle |Z(\beta)|^{2n} \rangle$.

These are quenched averages. Sometimes, it happens that quenched and annealed averages to leading order coincide $\langle Z^n \rangle = \langle Z \rangle^n$ in the large $N$ limit. This means that replicas $1,...,n$ are uncoupled, but in this case it also means that one may neglect the fluctuations of level densities: only average densities matter. It may be also true that, again, to leading order $\langle ZZ^* \rangle = \langle Z \rangle \langle Z^* \rangle$: this is the situation in phase I.

In phase II the situation is the same, provided one is content with the evaluation of $\log Z$ or the energy density. The latter is frozen at the lowest value $-e_c$. If we wish instead to get into the details of the actual distribution of overlaps, we need to understand how the measure is split between the infinity of levels having energy density $-e_c + O(1/N)$, in particular the two

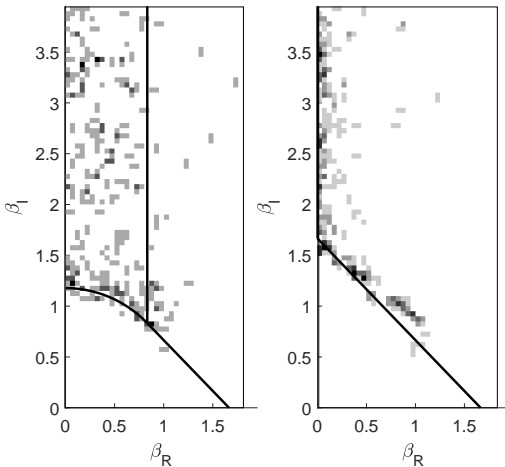

Figure 6: Random energy model zeroes without (left) and with (right) level-repulsion. Dark areas show the locations of zeros in a numerical calculation with $N = 14$. The position of zeros was obtained by partitioning the complex $(\beta_R, \beta_I)$ plane into small rectangles, of a size small compared to the typical distance between zeros at the simulated $N$. A zero inside a rectangle was identified by the residue theorem, by integrating over $1/Z$ (equivalently, checking if the complex angle of $\log(Z)$ changed when traversing the boundary).

lowest. This information, which we shall not discuss here, needs [22] that one considers the quenched average, break replica symmetry, and for this use the correlation between levels.

In phase III the situation is hybrid: the limit $\frac{1}{N}\langle \log Z(\beta) \rangle$ simply does not exist. On the other hand, $\frac{1}{N}\langle \log |Z(\beta)| \rangle$ does, and furthermore $\frac{1}{N}\langle |Z(\beta)|^{2n} \rangle = \frac{1}{N}\langle |Z(\beta)|^2 \rangle^n$. Here the replicas corresponding to $Z$ and $Z^*$ are coupled, but different pairs may or may not be, in this case they are not. At any rate, the correlation of level densities enters into the solution.

## 4 Model with level repulsion

In Figure 6 we show the empirical density of zeroes obtained in simulations with the REM (left), and those of the same density but with level repulsion added (right). Let us discuss this case in detail. In Figure 7 we show the SFF for the same model.

We start with a set set of $2^N$ levels $u_i$, with level repulsion, distributed uniformly over $(-1, 1)$. This is done by first generating a sample from a spectrum of a GOE random matrix (sampled efficiently as described in [32]). A function $\lambda \to g(\lambda)$ is applied to the eigenvalues, chosen to obtain a uniform density. We then transform again, to obtain any desired energy density:

$$E_i = \Phi^{-1}(u_i), \qquad u_i = \Phi(E_i). \tag{18}$$

Two features concerning the density of levels $\langle \rho(Ne) \rangle = e^{Ns(e)}$ of the new model are important:

- Zero-temperature entropy finite or zero: $\lim_{T \to 0} \lim_{N \to \infty} s(e) = s_o$.

- Freezing transition temperature: $\frac{1}{T_c} = \beta_c = \lim_{e \to e_c} \lim_{N \to \infty} \frac{\partial s}{\partial e}$ finite or infinite.

If we wish to mimic the SYK model we need $\beta_c = \infty$ and $s_o > 0$, while a 'quantum REM' (and other spin-glasses) would have $\beta_c$ finite and $s_o = 0$. For the latter case, $\Phi$ is the error function,

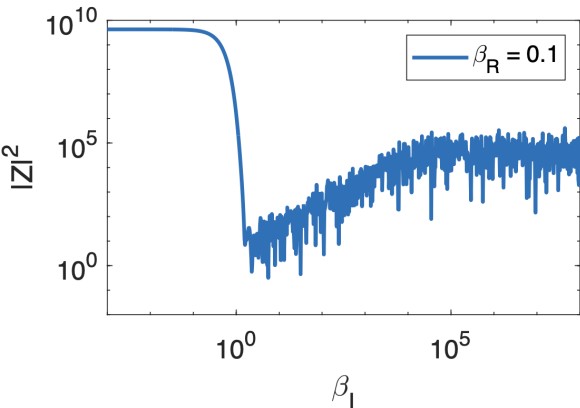

Figure 7: Spectral form factor of the model with a Gaussian distribution of levels like the REM, but with level repulsion.

so we have a REM level density with level repulsion.

$$\langle \tilde{\rho}(E)\rangle \mathrm{d}E = \text{uniform} \times \frac{\mathrm{d}u}{\mathrm{d}E}\mathrm{d}E = \mathcal{N}e^{-\frac{E^2}{N}}\,, \tag{19}$$

$$\frac{\mathrm{d}\Phi}{\mathrm{d}E} = \langle \tilde{\rho}(E)\rangle\,. \tag{20}$$

We define $\rho(E) = 2^N \tilde{\rho}(E)$ normalized as before, $\int \mathrm{d}\langle \rho(E)\rangle = 2^N$. The effect of the level repulsion in manifest in the connected correlation between energy levels:

$$\langle \delta\rho(u)\delta\rho(u')\rangle = \langle \rho(u)\rho(u')\rangle - \langle \rho(u)\rangle\langle \rho(u')\rangle$$
$$= 2^{2N}\left\{R\left(2^N(u-u')\right)-1\right\}\,, \tag{21}$$

with

$$R(s)-1 = \delta(s) - \frac{\sin^2(\pi s)}{(\pi s)^2}\,. \tag{22}$$

Here and in what follows for the purposes of analytic computations we will use the GUE statistics, which is slightly simpler and does not change significantly the discussion. We shall need the Fourier version:

$$\int \mathrm{d}s\, e^{ius}\,[R(s)-1] = g(u)\,, \tag{23}$$

and

$$g(u) = \begin{cases} |u|\,, & \text{for } |u| < 2\pi\,, \\ \\ 1\,, & \text{for } |u| > 2\pi\,. \end{cases} \tag{24}$$

We now make the non-linear transformation (18). The density of levels transform in the following way:

$$\rho(u) = \sum_i \delta(u-u_i) = 2^N \langle \rho(E)\rangle^{-1}\rho(E)\,, \tag{25}$$

and the expression for the correlations becomes:

$$\langle \delta\rho(E)\delta\rho(E')\rangle = \langle \rho(E)\rho(E')\rangle - \langle \rho(E)\rangle\langle \rho(E')\rangle$$
$$= \langle \rho(E)\rangle\langle \rho(E')\rangle\left[R(2^N(\Phi(E)-\Phi(E')))-1\right]\,. \tag{26}$$

We can evaluate the contribution to the SFF which comes from connected correlations. The computation is very close to the one in [2]:

$$\langle |Z(\beta)|^2\rangle = \int dw de de' g(w) \langle \rho(Ne)\rangle e^{iw2^N\Phi(Ne)-\beta Ne} \langle \rho(Ne')\rangle \, e^{-iw2^N\Phi(Ne')-\beta^* Ne'}$$

$$= \int dw de de' g(w) \langle \rho(Ne)\rangle \langle \rho(Ne')\rangle e^{iw2^N(\Phi(Ne)-\Phi(Ne'))-\beta_R N(e+e')-i\beta_I N(e-e')}. \quad (27)$$

We define $e_+ = \frac{1}{2}(e+e')$ and $e_- = (e-e')$ and we make the following expansion:

$$2^N(\Phi(E)-\Phi(E')) = \frac{N}{2} 2^N \partial_E \Phi(E)|_{E=E_+=Ne_+}(e-e'). \quad (28)$$

Considering first the exponentially large terms, we have

$$\int de_- \, e^{iN\left(\frac{1}{2}w2^N\partial_E\Phi(E)|_{E=E_+=Ne_+}-\beta_I\right)e_-} \to \delta\left(\frac{w}{2}\langle\rho(Ne_+)\rangle - \beta_I\right), \quad (29)$$

where we used that $2^N \partial_E \Phi(E)|_{E=E_+=Ne_+} = \langle \rho(Ne_+)\rangle$. Plugging this in Eq. (27) we have:

$$\langle |Z(\beta)|^2\rangle = \int dw de_+ g(w)\delta\left(\frac{w}{2}\langle\rho(Ne_+)\rangle - \beta_I\right)\langle\rho(Ne_+)\rangle^2 e^{-2\beta_R Ne_+} \quad (30)$$

$$= \int de_+ g\left(\frac{2\beta_I}{\langle\rho(Ne_+)\rangle}\right)\langle\rho(Ne_+)\rangle e^{-2\beta_R Ne_+}. \quad (31)$$

This integral represents an effective problem with density of states $g\left(\frac{2\beta_I}{\langle\rho(Ne_+)\rangle}\right)\langle\rho(Ne_+)\rangle$ at inverse temperature $2\beta_R$. The situation is represented in Figures 8 and 9. Below we discuss the result of the integral for different values of $\beta = \beta_R + i\beta_I$.

**Glassy phase**

This phase corresponds to phase II of the Poissonian REM, and does not exist in a model like SYK which has a transition only at infinite $\beta_R$. Above $\beta_R > \sqrt{\log 2}$ (and for sufficiently large $\beta_I$) for the modified REM the solution of the plateau reads:

$$\frac{1}{N}\log|Z(\beta)| = \beta_R e_c. \quad (32)$$

This is the same free energy as in the glassy phase.

**Liquid phase ('the slope')**

There is a liquid phase which is the same as phase I of the Poissonian REM, since this depends only on the level density and not the correlations. It corresponds to the 'slope' part of the SFF and its free energy reads:

$$f_I(\beta_R, \beta_I) = \lim_{N\to\infty} \frac{1}{N}\log|Z| = \log 2 + \frac{1}{4}(\beta_R^2 - \beta_I^2). \quad (33)$$

**The regions with zeroes**

Just as in the case of the REM, we need to compare the analytic continuations to the 'incoherent' part, by calculating $\langle |Z(\beta)|^2\rangle$. There are several possibilities:

**(i) Plateau regime**

If $\beta_I > \pi\langle\rho(E=0)\rangle$, above the bell shape in Figs 8 and 9, then the entire integral is with $w$ in the plateau value, we have in fact $g(w) = 1$ for all energies and the integral becomes

$$\langle|Z(\beta)|^2\rangle = \int de_+ \langle\rho(e_+)\rangle e^{-2\beta_R N e_+} = Z(2\beta_R). \tag{34}$$

The value of $e_+$ that dominates is $e(2\beta_R)$. This is the same result as in the REM, and (for sufficiently low $\beta_R$) we have a finite density of zeros. It is easy to convince oneself that in a plateau regime all models should have Fisher zeros, unless the dependence on $\beta_R$ is linear.

**(ii) Ramp I regime**

Here $\beta_I$ is such that the cycle of $e^{i\beta_I E}$ is longer than the level spacing at the edge of the spectrum - the largest of all level spacings - $\beta_I < \pi\langle\rho(e_c)\rangle$.[3] In this regime we have $g(w) = w$ for all energies and the integral becomes

$$\langle|Z|^2\rangle = \beta_I \int_{-e_c}^{e_c} de_+ e^{-2\beta_R N e_+} = \beta_I \frac{1}{N\beta_R} \sinh(2N\beta_R e_c),$$

$$\frac{1}{N}\ln|Z(\beta_R,\beta_I)| = \frac{1}{2N}\ln\beta_I + \frac{1}{2N}\ln|\sinh(2N\beta_R e_c)| - \frac{1}{2N}\ln|\beta_R| - \frac{1}{2N}\ln N. \tag{35}$$

**(iii) Ramp II regime**

The transition between the ramp and the plateau occurs in the intermediate regime where $\pi\langle\rho(E_c)\rangle < \beta_I < \pi\langle\rho(E=0)\rangle$ (see Figs 8 and 9).

We must integrate two regimes of $w$: *i)* $\beta_I < \pi\langle\rho(E)\rangle$ then $|w| < 2\pi$ and *ii)* $\beta_I > \pi\langle\rho(E)\rangle$ and then $|w| > 2\pi$.

$$\int_{-e_c}^{-e^*} de_+ \langle\rho(e_+)\rangle e^{-2\beta_R N e_+} + \int_{-e^*}^{e^*} de_+ w \langle\rho(e_+)\rangle e^{-2\beta_R N e_+} + \int_{e^*}^{e_c} de_+ \langle\rho(e_+)\rangle e^{-2\beta_R N e_+}, \tag{36}$$

where $\pi\langle\rho(\pm e^*)\rangle = \beta_I$ are the $w = 2\pi$ intercepts. This gives:

$$\int_{-e_c}^{-e^*} de \langle\rho(e)\rangle e^{-2\beta_R N e_+} + \int_{e^*}^{e_c} de \langle\rho(e)\rangle e^{-2\beta_R N e_+} + \beta_I \int_{-e^*}^{e^*} de\; e^{-2\beta_R N e_+}. \tag{37}$$

Eq (37) may be understood better by rewriting it as:

$$\int_{-e_c}^{e_c} de\; e^{Ns(e)} e^{-2\beta_R N e}, \tag{38}$$

where

$$s(e) = \begin{cases} \frac{1}{N}\ln\rho(e), & \pi\rho(e) < \beta_I, \\ \frac{1}{N}\ln\beta_I, & \pi\rho(e) > \beta_I. \end{cases} \tag{39}$$

---

[3]This condition is quite different if there is non-zero entropy at zero temperature or not. The largest level spacing in the REM is of order one in energy, while for SYK it is much smaller.

For every value of $\beta_I$ the saddle point evaluation may be viewed in Figs 8 and 9.

- for $2\beta_R > \beta_R^{max}$ the integral is dominated by the lowest energy $e_+ = -e_c$.

- for $\beta_R^{max} > 2\beta_R > \beta_R^{min}$ the integral is dominated by the saddle $e_+ = e(2\beta_R)$, that is the same as it would be if there would not be a truncation, i.e. for large $\beta_I$. This regime corresponds to the plateau.

- For $2\beta_R < \beta_R^{min}$ the integral is frozen around the intersection with the curve $w = 2\pi$. Its value describes the ramp and it is given by the following contributions:

$$\langle |Z(\beta)|^2 \rangle \simeq I_1 + I_2 \simeq \beta_I \int de \underbrace{\left[ \theta(-e^* - e)e^{N\beta_{min}(e+e^*)} + \theta(e + e^*) \right] e^{-N2\beta_R e}}_{e^{\mu(e)}}, \quad (40)$$

where $\theta(x)$ is the step function, so that:

$$I_1 = \int_{-e_c}^{-e^*} de_+ \langle \rho(Ne_+) \rangle e^{-2\beta_R Ne_+} \quad (41)$$

$$\simeq e^{Ns(-e^*)+2\beta_R Ne^*} \int_{-e_c}^{-e^*} de\, e^{N\left(\beta_R^{min}(e+e^*)-2\beta_R(e+e^*)\right)} \simeq \beta_I e^{2\beta_R Ne^*} \frac{1}{N} \frac{1}{\beta_R^{min} - 2\beta_R}, \quad (42)$$

where we used that $e^{Ns(-e^*)} = \beta_I$ and

$$I_2 = \beta_I \int_{-e^*}^{e_*} de_+ e^{-2\beta_R Ne_+} = \beta_I \frac{1}{2\beta_R N} \sinh(2\beta_R Ne^*) \simeq \beta_I \frac{1}{N} \frac{1}{4\beta_R} e^{2\beta_R Ne^*}. \quad (43)$$

Note that $\beta_R^{min}$ is a function of $\beta_I$. Altogether this regime describes the transition between the ramp and the plateau because increasing $\beta_I$ the singularity of the effective entropy $s(e)$ moves at higher energies (and thus lower $\beta_R^{min}$) and one crosses over from the situation where the saddle point is frozen at the singularity (ramp) to the one in which it occurs at the energy $e(2\beta_R)$ (plateau), see Fig. 9. The regime in which Eq. (41) holds is for $2\beta_R < \beta_R^{min}$. The point where $2\beta_R = \beta_R^{min}$ where there is an apparent divergence should be treated separately and represents the trasition point between ramp and plateau.

**Transition**

The transition between these regimes is obtained by comparing the magnitude of free energies. Since $e^* \sim e_c$ in this regime, we may estimate:

$$\lim_{N\to\infty} \frac{1}{N} \log |Z| = \beta_R e_c, \quad (44)$$

namely the same free energy as in the glassy phase. This should be therefore compared with $f_I$:

$$\beta_I = \sqrt{\beta_R^2 - 4\beta_R\sqrt{\log 2} + 4\log 2} = 2\sqrt{\log 2} - \beta_R, \quad (45)$$

where we used $e_c = \sqrt{\log 2}$. This is the line we see in Figure 6 (left) that crosses the real axes at $\beta_c = 2\sqrt{\log 2}$, the critical temperature of the REM.

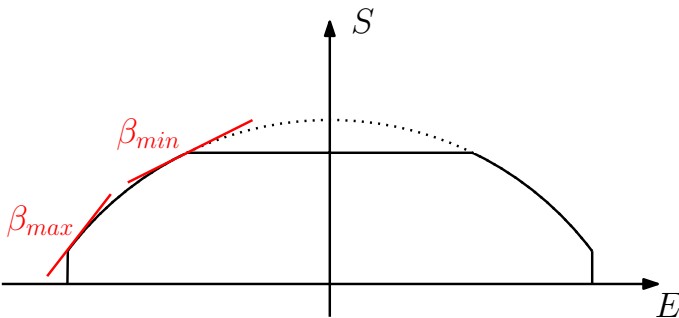

Figure 8: Equation (37) may be seen as the partition function associated to an entropy with this shape: a truncated density level, with truncation of height $\ln \beta_I$.
• If $\ln \beta_I < S(e \to 0)$ the saddle point equation determines two temperatures: $\beta_R^{max}$ where the saddle freezes in the lower energy (and finite entropy, if there is one), and $\beta_R^{min}$ (a function of $\beta_I$) where the saddle freezes on the intersection of the bell shape with the flat part. In the latter, the value of the integral is proportional to $\beta_I$. The value $\beta_c$ may be infinite, as is in SYK.

## Density of zeroes

In the **plateau regime**, $2 \ln |Z(\beta)|$ is the same as one would obtain for real temperature $2\beta_R$.

The density of zeroes is then:

$$\Omega(\beta_R, \beta_I) = \frac{1}{4\pi N} \frac{\partial^2 \ln Z(2\beta_R)}{\partial \beta_R^2} \frac{N}{\pi} \left[ \langle e^2 \rangle_{2\beta_R} - \langle e \rangle_{2\beta_R}^2 \right] = O(1) > 0, \tag{46}$$

where averages are over a real Gibbs measure.

In the **Ramp I** regime the density of zeros is:

$$\Omega \propto \nabla^2 \frac{1}{N} \log |Z| = \frac{1}{2N} \left( \frac{1}{\beta_R^2} - \frac{1}{\beta_I^2} \right) - \frac{2 e_c^2 N}{\sinh^2(2N\beta_R e_c)}$$

$$= 2 e_c \times \underbrace{N e_c \left\{ \left( \frac{1}{(2N e_c \beta_R)^2} \right) - \frac{1}{\sinh^2(2N\beta_R e_c)} \right\}} - \frac{1}{2N\beta_I^2}, \tag{47}$$

the underlined term has integral $O(1)$ and 'fat' tails $\propto \frac{1}{N\beta_R^2}$. In the **Ramp II** regime we need to expand around the intersection point of the $S(-e^*) = \ln \beta_I$. This leads to:

$$\Omega(\beta_R, \beta_I) = \frac{1}{2\pi N} \left[ \frac{\partial^2 \ln |Z|}{\partial \beta_R^2} + \frac{\partial^2 \ln |Z|}{\partial \beta_I^2} \right] \tag{48}$$

$$= \frac{1}{2\pi} \left[ 4N(\langle e^2 \rangle_\mu - \langle e \rangle_\mu^2) + \left\langle \frac{\partial^2 \mu}{\partial \beta_I^2} + \left( \frac{\partial \mu}{\partial \beta_I} \right)^2 \right\rangle_\mu - \left\langle \frac{\partial \mu}{\partial \beta_I} \right\rangle_\mu^2 - \frac{1}{N} \frac{1}{\beta_I^2} \right] \tag{49}$$

$$\sim \frac{2N}{\pi} (\langle e^2 \rangle_\mu - \langle e \rangle_\mu^2) \sim O\left( \frac{1}{N} \right), \tag{50}$$

where $\langle \bullet \rangle_\mu = \frac{\int de \, \bullet \, e^{\mu(e)}}{\int de \, e^{\mu(e)}}$, and we have neglected the derivatives with respect to $\beta_I$ because this regime concerns exponentially large $\beta_I$.

Note that the ramp has a smaller density of zeroes than the plateau (cf also Fig 6). This however affects only logarithmically the large deviation function for the largest spike.

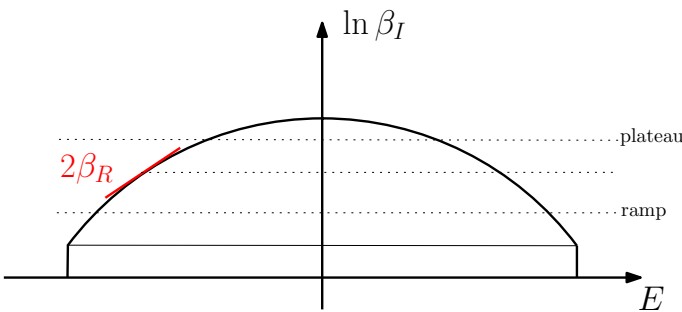

Figure 9: The transition from ramp to plateau upon increasing the level of $\ln \beta_I$, indicated with a dashed line. Given a value of $\beta_R$, we have the saddle point where the gradient is $2\beta_R$. (a) If $\ln \beta_I$ is above this value, then the saddle dominates and the result is independent of $\beta_I$: this is the plateau. (b) If it is below this value, the integral is dominated by the dashed line level: this is the ramp. (c) For lower values of $\ln \beta_I$, the lowest energy density states dominate.

## 5 Conclusions

The problem of the spectral form factor is one of computing a free energy in the complex temperature plane, with the very important caveat that, unlike the situation in thermodynamics, one goes to complex inverse temperatures $|\beta|$ which do not remain constant in the thermodynamic limit. The computation of fluctuations of the free energy sampled over ranges of $|\beta|$ that are exponential in the size $N$ is thus a problem of Large Deviation theory. Note that the fact that the larger spikes are exponentially rare and exponentially thin (in $N$) implies that the effect of these fluctuations will be ignored by higher moments of the trace, i.e. by a replica treatment of the problem.

The form factor may be expressed as a sum of the time-correlations $\sum_n \langle A_n(t)A_n(0) \rangle$ of an exponential set of operators [33]. It would be interesting to see if the zeroes move chaotically with the addition of each new term, even for a single sample.

We have given an estimate of the size of the largest deviations (spikes) assuming that the system may be considered for these purposes to be featureless, i.e. to have level correlations given only by the standard level repulsion. That this assumption is valid for *all* large fluctuations of SYK is plausible, but not given.

The 'gravity' side of SYK would give us the probability density of large spikes of a single sample at a given time through the density of Fisher zeroes obtained through the Laplacian in temperature, but the actual position would require exponential precision in $N$. Here we have only discussed the largest spikes. It seems possible that a more detailed distribution of fluctuations may be obtained directly from the theory of random polynomials.

## Acknowledgements

**Funding information**  This work was supported by 'Investissements d'Avenir' LabEx PALM (ANR-10-LABX-0039-PALM) (EquiDystant project, L. Foini) and Simons Foundation Grant No 454943 (J. Kurchan).

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
