# Peer review of "Fisher zeroes and the fluctuations of the spectral form factor of chaotic systems"

_SciPost Physics, doi:SciPost Phys. 17, 114 (2024)_

## Round 2 · Referee Report · Anonymous (Referee 1) · 2023-6-2

Strengths

1-Authors point out the interesting connection between the Fisher zeros and different regimes of the spectral form factor, which is of great interests in condensed matter and high energy physics.
2-Authors propose new analytical tractable model with energy repulsion by a transformation of the GOE spectral.

Weaknesses

1- Authors focus on random-matrix-like toy models. There are no discussions on systems with physical Hamiltonian, such as the SYK model or spin chains. 2-Authors should improve the language and formatting.

Report

In this paper, authors study the spectral form factor (SFF) of two toy models: the random energy model with or without level repulsions. Using these models, authors explain the relation between ramp/plateau behaviors in the SFF and the existence of Fisher zeros of partition functions. I find the topic of this paper are appropriate for SciPost Physics, and the results are of high originality. I suggest the publication of this work after minor revision.

Requested changes

Include discussions on physical Hamiltonians, correct grammar errors, and improve the formatting of equations.

  • validity: high
  • significance: good
  • originality: high
  • clarity: ok
  • formatting: acceptable
  • grammar: reasonable

Author:  Jorge Kurchan  on 2023-12-22  [id 4207]

(in reply to Report 1 on 2023-06-02)

Reply to the referees' reports

We thank both referees for the appreciation of our work. We also thank them for their suggestions which allow us to improve the presentation. Below we answer to their comments.

Referee 2

There are various instances where the manuscript is not self-contained and where the reader would benefit from more detail and clarification. For instance, the SYK model is mentioned in various places, and numerical data is presented. Yet, the model is not defined nor > described anywhere in the paper. Moreover, when describing the phase diagram of the random-energy model, the authors in some places refer to the absence of some phases in the SYK model.

We have included the definition

Related to the previous point: the relation between the different models is unclear: is the random-energy model supposed to be a toy model to describe the SFF of the SYK model?

No. The random energy model share in common only the region with a plateau. SYK as in general many-body systems have level repulsion and their SFF a has a ramp. For this reason we have studied the modification of the REM adding level repulsion. The peculiarly of the SYK of having chaotic behavior also at very low energies is not captured by our model, but we expect the phenomenon of the spikes to be the same.}

The authors argue that the sharp dips in the SFF are caused by near-misses of zeroes. This appears to be the main result of the paper. While this statement is not entirely surprising...

...perhaps. But please note that this picture immediately implies that these large fluctuations will not be described by averages over higher powers of the trace, i.e. they will not be captured by a description with finite number of replicas, since the spikes are thin and rare, and have negligible weight. We have added a phrase to that effect: `` Note that the fact that the larger spikes are exponentially rare and exponentially thin (in N) implies that the effect of these fluctuations will be ignored by higher moments of the trace, i.e. by a replica treatment of the problem."

... , I agree that it is interesting and perhaps not fully appreciated in the community. However, it is not clear to me how general this statement is: (i) can all dips in the SFF be described by this mechanism? (ii) Does this mechanism also apply to other types of models, other than SYK or the random-energy model, e.g., kicked-Ising type models?

For Hamiltonian systems we expect the mechanism to be the same. However it is not clear how to generlise this to non-equilibrium set-ups. The spikes could still be zeros of the SFF in time, even if this is not the imaginary part of a complex temperature.}

Another way to understand the SFF is in terms of correlation functions of Pauli operators: \sum tr(P_k P_k(t)) where the sum runs over all 4^n Pauli operators P_k in a system of n qubits (see, e.g., 1706.05400, Sec. 3). Can the authors see a connection between their results and these correlation functions of Pauli strings?

This raises the interesting possibility that each subsequent term in the correlation shifts the zeroes chaotically. We have now mentioned this: `` The form factor may be expressed as a sum of the time-correlations \sum_n \langle A_n(t) A_n(0)\rangle of an exponential set of operators [.]. It would be interesting to see if the zeroes move chaotically with the addition of each new term, even for a single sample. ".}

There also exist examples where the fluctuations of the SFF don't vanish even when averaged over an ensemble of systems, for instance in Clifford Floquet circuits (see e.g., 2210.10129). Can the authors comment on whether these types of fluctuations (or their density) are >qualitatively different than the ones studied here?

We should understand this example better, anyway Floquet systems are beyond the present study which studies partition functions in complex temperature.

As far as I understand, Section 3 on the Poissonian random-energy model is just a summary of known results in the literature. Yet, it is in some places written as if the authors present their own results ("we", "our", ...) . The authors should be more careful to separate their original findings from existing results.

\f We have made this more clear.

I have to say that, in my opinion, the figures are of subpar quality. Examples: Fig. 1: legend should be "\beta_R". Fig. 2: why "d" and not "a" as in Eq. (3)? Fig. 4: no axis label, values at phase boundaries would be helpful. It should also be mentioned that this is not their result > >but reproduced from the literature. Figure 7 is never mentioned anywhere in the text. Figs. 8 and 9 are in my opinion not comprehensible and below publication threshold. Among other things, what is the meaning of the dotted, thin, and thick blue lines in Fig. 8?

We have improved the figures.

Overall, it feels that the paper is not prepared very carefully (see previous comment on figures). There also are a number of typos throughout the text. Besides minor typos, most strikingly, in various instances, the authors write "spectral factor" instead of "spectral form factor". >The alignment of multi-line equations can be improved in some places. Equations should have commas or periods at the end of the line.

f We have corrected the typos.

I would argue that the notion of "spike" is usually associated with a sharp increase of some quantity. In contrast, the SFF rather has "sharp dips".

This is true, but we hope it is sufficiently clear from the results.

What is G_\infty in Eq. (3)?

We have clarified it below Eq. 3.

Referee 1

Include discussions on physical Hamiltonians, correct grammar errors, and improve the formatting of equations.

We have done this, see other report.

Author:  Jorge Kurchan  on 2023-12-22  [id 4206]

(in reply to Report 1 on 2023-06-02)
Category:
remark

Reply to the referees' reports

We thank both referees for the appreciation of our work. We also thank them for their suggestions which allow us to improve the presentation. Below we answer to their comments.

Referee 2

There are various instances where the manuscript is not self-contained and where the reader would benefit from more detail and clarification. For instance, the SYK model is mentioned in various places, and numerical data is presented. Yet, the model is not defined nor > described anywhere in the paper. Moreover, when describing the phase diagram of the random-energy model, the authors in some places refer to the absence of some phases in the SYK model.

We have included the definition

Related to the previous point: the relation between the different models is unclear: is the random-energy model supposed to be a toy model to describe the SFF of the SYK model?

No. The random energy model share in common only the region with a plateau. SYK as in general many-body systems have level repulsion and their SFF a has a ramp. For this reason we have studied the modification of the REM adding level repulsion. The peculiarly of the SYK of having chaotic behavior also at very low energies is not captured by our model, but we expect the phenomenon of the spikes to be the same.}

The authors argue that the sharp dips in the SFF are caused by near-misses of zeroes. This appears to be the main result of the paper. While this statement is not entirely surprising...

...perhaps. But please note that this picture immediately implies that these large fluctuations will not be described by averages over higher powers of the trace, i.e. they will not be captured by a description with finite number of replicas, since the spikes are thin and rare, and have negligible weight. We have added a phrase to that effect: `` Note that the fact that the larger spikes are exponentially rare and exponentially thin (in N) implies that the effect of these fluctuations will be ignored by higher moments of the trace, i.e. by a replica treatment of the problem."

... , I agree that it is interesting and perhaps not fully appreciated in the community. However, it is not clear to me how general this statement is: (i) can all dips in the SFF be described by this mechanism? (ii) Does this mechanism also apply to other types of models, other than SYK or the random-energy model, e.g., kicked-Ising type models?

For Hamiltonian systems we expect the mechanism to be the same. However it is not clear how to generlise this to non-equilibrium set-ups. The spikes could still be zeros of the SFF in time, even if this is not the imaginary part of a complex temperature.}

Another way to understand the SFF is in terms of correlation functions of Pauli operators: \sum tr(P_k P_k(t)) where the sum runs over all 4^n Pauli operators P_k in a system of n qubits (see, e.g., 1706.05400, Sec. 3). Can the authors see a connection between their results and these correlation functions of Pauli strings?

This raises the interesting possibility that each subsequent term in the correlation shifts the zeroes chaotically. We have now mentioned this: `` The form factor may be expressed as a sum of the time-correlations \sum_n \langle A_n(t) A_n(0)\rangle of an exponential set of operators [.]. It would be interesting to see if the zeroes move chaotically with the addition of each new term, even for a single sample. ".}

There also exist examples where the fluctuations of the SFF don't vanish even when averaged over an ensemble of systems, for instance in Clifford Floquet circuits (see e.g., 2210.10129). Can the authors comment on whether these types of fluctuations (or their density) are >qualitatively different than the ones studied here?

We should understand this example better, anyway Floquet systems are beyond the present study which studies partition functions in complex temperature.

As far as I understand, Section 3 on the Poissonian random-energy model is just a summary of known results in the literature. Yet, it is in some places written as if the authors present their own results ("we", "our", ...) . The authors should be more careful to separate their original findings from existing results.

\f We have made this more clear.

I have to say that, in my opinion, the figures are of subpar quality. Examples: Fig. 1: legend should be "\beta_R". Fig. 2: why "d" and not "a" as in Eq. (3)? Fig. 4: no axis label, values at phase boundaries would be helpful. It should also be mentioned that this is not their result > >but reproduced from the literature. Figure 7 is never mentioned anywhere in the text. Figs. 8 and 9 are in my opinion not comprehensible and below publication threshold. Among other things, what is the meaning of the dotted, thin, and thick blue lines in Fig. 8?

We have improved the figures.

Overall, it feels that the paper is not prepared very carefully (see previous comment on figures). There also are a number of typos throughout the text. Besides minor typos, most strikingly, in various instances, the authors write "spectral factor" instead of "spectral form factor". >The alignment of multi-line equations can be improved in some places. Equations should have commas or periods at the end of the line.

f We have corrected the typos.

I would argue that the notion of "spike" is usually associated with a sharp increase of some quantity. In contrast, the SFF rather has "sharp dips".

This is true, but we hope it is sufficiently clear from the results.

What is G_\infty in Eq. (3)?

We have clarified it below Eq. 3.

Referee 1

Include discussions on physical Hamiltonians, correct grammar errors, and improve the formatting of equations.

We have done this, see other report.

---

## Round 2 · Referee Report · Anonymous (Referee 2) · 2023-6-23

Strengths

  • timely subject
  • interesting perspective on the spectral form factor
  • analytical results for a toy-model with level repulsion

Weaknesses

  • importance and generality of results unclear
  • manuscript not self-contained
  • below-average presentation

Report

The manuscript by Bunin, Foini, and Kurchan is concerned with the fluctuations of the spectral form factor which are present when considering single instances of quantum systems (as opposed to averaging over an ensemble of systems). The authors argue, and provide explicit numerical examples, that these fluctuations can be understood with respect to zeroes in the complex temperature plane when interpreting the SFF as the modulus of the partition function. A model of random energies with level repulsion is considered, for which the authors study the density of zeroes in different phases and obtain expressions for the averaged SFF in the slope, ramp, and plateau regime.

Overall, this is an interesting manuscript that contains results that can certainly be helpful to other researchers in the community. The spectral form factor is a timely quantity with connections to different subfields, including both theory and experiment. Moreover, very often it is only the averaged SFF that is studied in the literature, whereas the authors here provide an interesting perspective on individual realizations. I therefore believe that the manuscript definitely warrants publication in some form or another.

Having said this, I also think that in its current form, the manuscript is clearly not ready for publication. In my opinion, a number of revisions are necessary, both regarding the (generality of the) results themselves, as well as regarding the overall presentation of the paper. The following points should be taken into account be the authors' when preparing their resubmission (not necessarily in order of importance):

  • There are various instances where the manuscript is not self-contained and where the reader would benefit from more detail and clarification. For instance, the SYK model is mentioned in various places, and numerical data is presented. Yet, the model is not defined nor described anywhere in the paper. Moreover, when describing the phase diagram of the random-energy model, the authors in some places refer to the absence of some phases in the SYK model.

  • Related to the previous point: the relation between the different models is unclear: is the random-energy model supposed to be a toy model to describe the SFF of the SYK model?

  • The authors argue that the sharp dips in the SFF are caused by near-misses of zeroes. This appears to be the main result of the paper. While this statement is not entirely surprising, I agree that it is interesting and perhaps not fully appreciated in the community. However, it is not clear to me how general this statement is: (i) can all dips in the SFF be described by this mechanism? (ii) Does this mechanism also apply to other types of models, other than SYK or the random-energy model, e.g., kicked-Ising type models?

  • Another way to understand the SFF is in terms of correlation functions of Pauli operators: $SFF \sim \sum_k^{4^n} tr[P_k P_k(t)]$ where the sum runs over all $4^n$ Pauli operators $P_k$ in a system of $n$ qubits (see, e.g., 1706.05400, Sec. 3). Can the authors see a connection between their results and these correlation functions of Pauli strings?

  • There also exist examples where the fluctuations of the SFF don't vanish even when averaged over an ensemble of systems, for instance in Clifford Floquet circuits (see e.g., 2210.10129). Can the authors comment on whether these types of fluctuations (or their density) are qualitatively different than the ones studied here?

  • As far as I understand, Section 3 on the Poissonian random-energy model is just a summary of known results in the literature. Yet, it is in some places written as if the authors present their own results ("we", "our", ...) . The authors should be more careful to separate their original findings from existing results.

  • I have to say that, in my opinion, the figures are of subpar quality. Examples: Fig. 1: legend should be "$\beta_R$". Fig. 2: why "$d$" and not "$a$" as in Eq. (3)? Fig. 4: no axis label, values at phase boundaries would be helpful. It should also be mentioned that this is not their result but reproduced from the literature. Figure 7 is never mentioned anywhere in the text. Figs. 8 and 9 are in my opinion not comprehensible and below publication threshold. Among other things, what is the meaning of the dotted, thin, and thick blue lines in Fig. 8?

  • Overall, it feels that the paper is not prepared very carefully (see previous comment on figures). There also are a number of typos throughout the text. Besides minor typos, most strikingly, in various instances, the authors write "spectral factor" instead of "spectral form factor". The alignment of multi-line equations can be improved in some places. Equations should have commas or periods at the end of the line.

  • I would argue that the notion of "spike" is usually associated with a sharp increase of some quantity. In contrast, the SFF rather has "sharp dips".

  • What is $G_\infty$ in Eq. (3)?

Requested changes

See report.

  • validity: high
  • significance: good
  • originality: good
  • clarity: ok
  • formatting: acceptable
  • grammar: good

Author:  Jorge Kurchan  on 2023-09-22  [id 4002]

(in reply to Report 2 on 2023-06-23)

  • There are various instances where the manuscript is not self-contained and where the reader would benefit from more detail and clarification. For instance, the SYK model is mentioned in various places, and numerical data is presented. Yet, the model is not defined nor described anywhere in the paper. Moreover, when describing the phase diagram of the random-energy model, the authors in some places refer to the absence of some phases in the SYK model.

We have included the definition

  • Related to the previous point: the relation between the different models is unclear: is the random-energy model supposed to be a toy model to describe the SFF of the SYK model?

No. The random energy model share in common only the region with a plateau. SYK as in general many-body systems have level repulsion and their SFF a has a ramp. For this reason we have studied the modification of the REM adding level repulsion. The peculiarly of the SYK of having chaotic behavior also at very low energies is not captured by our model, but we expect the phenomenon of the spikes to be the same.

  • The authors argue that the sharp dips in the SFF are caused by near-misses of zeroes. This appears to be the main result of the paper. While this statement is not entirely surprising...

** ...perhaps. But please note that this picture immediately implies that these large fluctuations will not be described by averages over higher powers of the trace, i.e. they will not be captured by a description with finite number of replicas, since the spikes are thin and rare, and have negligible weight. We have added a phrase to that effect: "Note that the fact that the larger spikes are exponentially rare and exponentially thin (in $N$) implies that the effect of these fluctuations will be ignored by higher moments of the trace, i.e. by a replica treatment of the problem."**

  • ... , I agree that it is interesting and perhaps not fully appreciated in the community. However, it is not clear to me how general this statement is: (i) can all dips in the SFF be described by this mechanism? (ii) Does this mechanism also apply to other types of models, other than SYK or the random-energy model, e.g., kicked-Ising type models?

For Hamiltonian systems we expect the mechanism to be the same. However it is not clear how to generlise this to non-equilibrium set-ups. The spikes could still be zeros of the SFF in time, even if this is not the imaginary part of a complex temperature.

  • Another way to understand the SFF is in terms of correlation functions of Pauli operators: $\sum tr(P_k P_k(t))$ where the sum runs over all $4^n$ Pauli operators $P_k$ in a system of $n$ qubits (see, e.g., 1706.05400, Sec. 3). Can the authors see a connection between their results and these correlation functions of Pauli strings?

This raises the interesting possibility that each subsequent term in the correlation shifts the zeroes chaotically. We have now mentioned this: "The form factor may be expressed as a sum of the time-correlations $\sum_n \langle A_n(t) A_n(0)\rangle$ of an exponential set of operators [.]. It would be interesting to see if the zeroes move chaotically with the addition of each new term, even for a single sample."

  • There also exist examples where the fluctuations of the SFF don't vanish even when averaged over an ensemble of systems, for instance in Clifford Floquet circuits (see e.g., 2210.10129). Can the authors comment on whether these types of fluctuations (or their density) are qualitatively different than the ones studied here?

We should understand this example better, anyway Floquet systems are beyond the present study which studies partition functions in complex temperature.

  • As far as I understand, Section 3 on the Poissonian random-energy model is just a summary of known results in the literature. Yet, it is in some places written as if the authors present their own results ("we", "our", ...) . The authors should be more careful to separate their original findings from existing results.

We have made this more clear.

  • I have to say that, in my opinion, the figures are of subpar quality. Examples: Fig. 1: legend should be "$\beta_R$". Fig. 2: why "$d$" and not "$a$" as in Eq. (3)? Fig. 4: no axis label, values at phase boundaries would be helpful. It should also be mentioned that this is not their result but reproduced from the literature. Figure 7 is never mentioned anywhere in the text. Figs. 8 and 9 are in my opinion not comprehensible and below publication threshold. Among other things, what is the meaning of the dotted, thin, and thick blue lines in Fig. 8?

We have improved the figures.

  • Overall, it feels that the paper is not prepared very carefully (see previous comment on figures). There also are a number of typos throughout the text. Besides minor typos, most strikingly, in various instances, the authors write "spectral factor" instead of "spectral form factor". The alignment of multi-line equations can be improved in some places. Equations should have commas or periods at the end of the line.

We have corrected the typos.

  • I would argue that the notion of "spike" is usually associated with a sharp increase of some quantity. In contrast, the SFF rather has "sharp dips".

This is true, but we hope it is sufficiently clear from the results.

  • What is $G_\infty$ in Eq. (3)?

We have clarified it below Eq. 3.

Referee 1

  • Include discussions on physical Hamiltonians, correct grammar errors, and improve the formatting of equations.

We have done this, see other report.

---

## Round 3 · Referee Report · Anonymous (Referee 1) · 2024-3-18

Report

The authors have improved the manuscript by adding results for the SYK model. I find the revision satisfactory. Now, I believe the paper satisfy the acceptance criteria of Scipost Physics.

---

## Round 3 · List of Changes

We have clarified notation and improved the figures as requested by the referees

We have added a phrase :
`` Note that the fact that the larger spikes are exponentially rare and exponentially thin (in $N$) implies that the effect of these fluctuations will be ignored by higher moments of the trace, i.e. by a replica treatment of the problem."

We have now mentioned this:
`` The form factor may be expressed as a sum of the time-correlations $\sum_n \langle A_n(t) A_n(0)\rangle$ of an exponential set of
operators [.]. It would be interesting to see if the zeroes move chaotically
with the addition of each new term, even for a single sample. ".

---

## Editorial Decision

published